# School of Nursing Climate Commitment: Nursing Faculty Bring Climate to the Classroom

**DOI:** 10.3390/ijerph21050589

**Published:** 2024-05-03

**Authors:** Shanda Demorest, Cara Cook, Elizabeth Schenk, Lisa Whitfield Harris, Andrea Earley

**Affiliations:** 1Practice Greenhealth, Reston, VA 20190, USA; 2School of Nursing, University of Minnesota Twin Cities Campus, Minneapolis, MN 55455, USA; 3Alliance of Nurses for Healthy Environments, Mt. Rainier, MD 20712, USA; cara@envirn.org; 4Providence Health, Renton, WA 98057, USA; elizabeth.schenk@wsu.edu; 5College of Nursing, Washington State University Health Sciences Campus, Spokane, WA 99202, USA; 6Jefferson College of Nursing, Thomas Jefferson University Philadelphia Campus, Philadelphia, PA 19107, USA; lisa.whitfield-harris@jefferson.edu (L.W.H.); andrea.earley12@gmail.com (A.E.); 7Jersey Shore University Medical Center, Neptune Township, NJ 07753, USA

**Keywords:** climate change, nursing education, health care sustainability

## Abstract

In 2021, the American Association of Colleges of Nursing (AACN) added “the impact of climate change on environmental and population health” into The Essentials: Core Competencies for Professional Nursing Education. Presently, little guidance exists for nursing faculty new to climate education. The year prior, the Nurses Climate Challenge (NCC)—a campaign to educate 50,000 health professionals about health impacts of climate change—launched the School of Nursing Commitment through a series of focus groups and collaborative content development. With an aim of increasing access to knowledge and tools to support education about the health impacts of climate change, the NCC Commitment partners with nursing schools and provides a community of practice. Partner schools use NCC resources in courses and report the number of students educated. Within three years, 61 nursing schools in 30 states joined the Commitment. Participants included academic health centers, research institutions, multi-state schools, and small private colleges, and programs ranged from AD to PhD. Faculty (1) integrated resources into didactic and clinical settings, such as population or organ-system content, leadership, and policy; and (2) used resources to support assignments. In four years, faculty reported educating over 37,700 students, using NCC resources in 439 educational sessions. The Commitment may be valuable for faculty fulfilling AACN Essentials by bringing climate change to the classroom, community, and bedside. Furthermore, the Commitment may be a replicable model for health professional education and inspiring action on climate change.

## 1. Introduction

As awareness grows among the global scientific community of how climate change impacts human health, it is essential for nurses to lend their credibility and knowledge to meeting this challenge. The United Nations recognizes advancing climate action as one of the 17 goals to achieving the 2030 Agenda for Sustainable Development [1]. There are multiple examples of nursing leadership on climate change and health, as well as further opportunities for nurses to lead action on this important public health issue [2,3,4,5]. However, integration of climate change into the nursing curriculum remains an obstacle to adequately preparing nurses to address this issue within practice and other professional roles [3].

International and national nursing organizations have emphasized the inclusion of climate change and health content in nursing education in order to prepare for and respond to climate-related impacts [6,7,8]. Further, the revision of the American Association of Colleges of Nursing The Essentials: Core Competencies for Professional Nursing Education [9], provides explicit language for the inclusion of “the impact of climate change on environmental and population health” in nursing curricula.

While nursing organizations have expressed the importance of including this topical content into nursing education via position statements [6,7,8,9], the integration of content varies greatly depending on the institution [10]. There is a lack of formalized integration of climate change into nursing education, which creates barriers for faculty to effectively integrate content into didactic, lab, and clinical practicum nursing courses [11,12]. Further action on climate change has been cited as a challenge for the nursing profession due to conceptual models of health that focus on an individual rather than community [13]. There is a number of existing resources available to informally guide faculty efforts to include this content into all levels of nursing education. The Global Consortium on Climate and Health Education (GCCHE), housed at Columbia University, provides open-access climate and health resources, such as syllabi, program plans, and sample slides for health professional schools globally to utilize [14]. In addition, the nurse-led Center for Climate Change, Climate Justice, and Health at the MGH Institute of Health Professions (MGHIHP), the Alliance of Nurses for Healthy Environments (ANHE), and nursing faculty across the U.S. have provided recommendations for nursing curriculum insertion points [11,15,16,17].

Further, the audience of our future health professionals appears ripe. Growing data suggest that nursing students are increasingly aware of the health impacts of climate change [12,18]. In a study of 280 health professional students by Ryan, Dubrow, and Sherman, 90% of participants agreed with the sentiment that health professionals have the responsibility to conserve resources in the workplace [19]. In a review of 32 international studies, authors reported that, of the nursing students and educators who were asked if climate and sustainability content should be included in nursing curricula, 100% responded positively [18]. Not only are future clinicians eager for this knowledge; current practicing health professionals are also keen to work for organizations that address climate action. In a 2024 study of 1,001 U.S. clinicians by The Commonwealth Fund, authors found that approximately 80% of clinicians reported they believed it was important for their health system to address climate change in support of their organizational mission, and approximately 75% of respondents shared they felt it was important that they personal reduced their environmental impact at work as health professionals [20]. Approximately 60% of respondents noted that the extent to which a future health system employer addressed climate change would have an influence on whether they applied for a job at that organization [20].

## 2. Knowledge and Action Framework

As health professional education gains momentum around climate and health, practice-oriented climate education for nurses globally is still in its early stages [12]. The primary foci of programs such as the GCCHE and MGHIHP are to increase knowledge of how human health is negatively impacted by climate change, such as acute effects from extreme heat, exacerbations in chronic cardiovascular and respiratory diseases, increased vector-borne illness such as malaria and Lyme disease, and mental health impacts, among others [13,14]. In 2018, the Nurses Climate Challenge (NCC) was developed to fill a gap in educational resources for nurses that could be applied in a variety of nursing practice settings in the U.S. [21]. The NCC was developed by Health Care Without Harm [22] & the Alliance of Nurses for Healthy Environments [23]—both non-profit organizations with missions to reduce human health harms from the environment. The novel concept introduced by the NCC involved preparing individuals to harness their newly acquired climate and health knowledge to take the next step towards health promotion: climate-change mitigation.

The NCC was created utilizing a backward design educational framework [24,25], which involved working backward from the ultimate goal of health promotion via climate mitigation as a desired result. In the U.S., health care service delivery contributes to approximately 8.5% of greenhouse gas emissions [26]; thus, a critical aspect of health promotion that nurses must take leadership in involves reducing greenhouse gas emissions. The backward design educational framework posits that learners must be presented with acceptable evidence to influence action which leads to the desired result [25].

In the process of developing materials, the project team engaged in a comprehensive literature and resource review of publicly-accessible materials on the health impacts of climate change, ranging from the National Climate Assessment [27,28] to the Centers for Disease Control [29] to state-based public health departments [30,31,32]. In addition, the project team investigated best practices for climate content delivery, drawing from exemplars such as The Climate Reality Project [33] and early-adopting faculty members’ experiences deploying climate content in the classrooms [5,17]. Finally, the nearly-four decades of collective history and experience of the program’s co-sponsors, Health Care Without Harm and the Alliance of Nurses for Healthy Environments, provided successful examples of education, mobilization, and advocacy [22,23].

The resulting NCC materials presented evidence designed to propel action, such as a basic orientation about climate science and resulting climatic impacts, an overview of health impacts experienced by individuals regionally throughout the U.S. (e.g., extreme heat, severe weather, air pollution, and vector borne disease, as identified by the American Public Health Association [34]), and health care’s contribution to the climate crisis [26].

To better prepare nurses to incorporate educational components into practice settings, the NCC team developed a variety of resources including PowerPoint presentations for educational sessions, template letters to engage institutional leadership, and resources for reducing the nurse’s environmental impact in the workplace (see Figure 1). These resources are housed on the NCC website (nursesclimatechallenge.org), and nurses can access these materials by registering to become a Nurses Climate Champion. Once nurses create an account, they are encouraged to educate others and track the number of health professional colleagues educated.

Inspired by the Snowflake Model developed by organizer Marshall Ganz [35,36], the NCC was crafted using a distributed leadership approach. The Snowflake Model empowers community members with knowledge and tools, which they can then use to enable others with that same knowledge, and so on (see Figure 2) [37].

By merging the Backward Design educational framework with the Snowflake Model, the exclusively nurse-developed NCC aimed to bring knowledge to a population via a distributed leadership model in the interest of increasing the moral and ethical impetus to take action to reduce the health effects of climate change. The goal of the NCC was to educate and inspire climate action in 50,000 health professionals by 2022.

## 3. Use of the Nurses Climate Challenge in the Classroom

One-hundred and eighty educational sessions were reported before the launch of the NCC SON Commitment by both academic and non-academic participants, reaching nearly 16,000 individuals from 2018 to 2019. During this period, a strong trend of utilization in academia appeared; nursing faculty reported educating 11,576 individuals in 81 educational sessions. Due to the robust response of nursing faculty utilizing the Challenge, the Challenge expanded in 2020 to include a School of Nursing (SON) Commitment with an aim of increasing access to knowledge and tools to support education about the health impacts of climate change and climate action in nursing education. The structure of the NCC involves inviting schools of nursing to join the commitment, use the NCC resources in courses, and report the number of students educated. A further aim of the commitment is to build momentum for climate education in nursing schools by publicly recognizing schools that are integrating climate and health content into their courses, and also to create a community of practice in which faculty share best practices to further inspire climate action in the nursing profession.

## 4. Methods

### 4.1. Focus Groups

Development of the SON Commitment began with two virtual focus groups of 12 faculty members total from 12 different universities with content expertise at the intersection of climate change and nursing education in late 2019. The NCC steering committee introduced the SON Commitment concept and presented faculty with draft commitment language, which included a description of the partnership, a summary of NCC resources, expectations of the NCC and the schools of nursing (see Table 1), and anticipated benefits for partners. Focus group participants were asked to provide feedback related to desirability, perceived value, feasibility of implementation, and missing components. Faculty feedback was incorporated into the development of the SON Commitment prior to the launch [38,39].

Focus group faculty consistently indicated that the implementation of the NCC SON Commitment would be undesirable if the partnership were to require an official contract. Universities across the country approach contracts in a variety of ways, which adds further barriers to including NCC content in nursing schools. Therefore, the commitment was developed without official contractual language or required signatures.

### 4.2. Phases of Recruitment

In 2020, the 12 universities with faculty representation in the focus groups were invited to officially participate as pilot SON Commitment partners, of which 11 gained leadership approval to participate (the reason for the single faculty member that did not receive approval was undisclosed). These faculty were asked to share about the Commitment and their experience in the SON Commitment within their climate and nursing education networks. In just over a year, 17 additional schools sought partnership without active recruitment by the NCC steering committee.

In May of 2021, a formalized recruitment process was established. Ongoing recruitment efforts began via webinars, newsletters, email, and social media in collaboration with multiple national faculty and student nursing organizations (e.g., the Association of Community Health Nursing Educators and National Student Nurses Association). Once a faculty member from a school expressed interest via the online interest form, a NCC steering committee member held a conversation with that faculty member to ensure that the NCC format and resources would benefit the school and the school of nursing anticipated being able to meet educational and communication expectations. In order to join and communicate the partnership publicly, nursing faculty obtained permission from their leadership (dean or associate dean) and, in some cases, their nursing school’s curriculum committee.

### 4.3. Community of Practice

As noted, nursing faculty encounter multiple barriers to integrating climate and health content into nursing curricula [11]; this was further validated by faculty in NCC convenings. To overcome the potential challenges faculty may face in integrating climate and health content into nursing curricula, the NCC committee established a community of practice that gathers virtually before spring and fall semesters and maintains electronic communication throughout the year. All SON Commitment faculty were invited to participate in the community of practice, which serves to share resources and best practices and provide a network of climate and nursing faculty allies. Multiple organic collaborations amongst faculty across universities have developed from this community of practice, such as serving as co-authors for climate and health publications or hosting Earth Day events for students.

### 4.4. Tracking Education

Upon educating health professional students, faculty submit a limited amount of data for each educational session, including the number of individuals educated, audience feedback, and follow-up actions pertaining to climate solutions that arose from the educational session. Data specific to the NCC Commitment is compiled quarterly to provide an analytics report for participating schools of nursing.

## 5. Results

### 5.1. Partner Schools and Implementation into Programs

Between January 2020 and December 2022, 61 schools of nursing in 30 states signed on to participate in the NCC SON Commitment [39]. Sixty of these schools were in the United States, and one was in Colombia. Program coordinators chose to include the Colombian university as a pilot international collaborator. Nursing programs in participating schools range from AD to PhD, and eight of the participating schools were faith-based. During that period, faculty in 47 of the 61 schools reported educating students with NCC resources (77%). To allow for flexibility, the Commitment does not require a specific date for content implementation after joining.

Many institutions used NCC resources in programming for multiple degrees. Seventy percent of participating institutions that have successfully implemented NCC resources into their SONs have done so in Bachelor of Science in nursing programs (see Figure 3). Six reporting faculty indicated using NCC resources in pre-licensure Masters of Science in nursing courses (12.7%), and nearly 20% of faculty have integrated NCC resources into advanced practice nursing or Doctor of Nursing Practice programs (see Figure 3). NCC resources were used least frequently in associate’s degree (three participants, comprising less than 6%) and PhD (one participant, comprising 2%) programs (see Figure 3). Seven participants implemented NCC resources into more than one degree program. As of December 2022, 14 schools (23%) had not reported educating students with NCC resources in any of their programs (see Figure 3).

### 5.2. Educational Sessions & Reach

In the first three years since the launch of the NCC SON Commitment, faculty in participating schools reported 439 sessions of education, representing 81% of all educational sessions of NCC materials during that period (see Figure 4). Faculty within participating schools reached 37,731 students between February 2020 and December 2022, with a median audience of 45 students per educational session (mean 86; range 1–3024) (See Figure 4).

Faculty participants were not required to report the type of course they implemented NCC materials into. However, the most common voluntarily reported courses included were:environmental healthcommunity or public healthclinical practicepopulation-specific (e.g., pediatrics)leadershippolicy and advocacyintroduction to the nursing professionsustainabilityglobal or planetary healthsocial determinants of healthdisaster nursingevidence-based practice or researchethics

Faculty in schools not participating in the Commitment after the program’s establishment reported 20 educational sessions, representing 3.6% of the overall total of educational sessions tracked from 2020–2022 (see Figure 4). These faculty reached 805 students in non-Commitment schools, with a median audience of 28 students per session (mean 40, range 1–120) (see Figure 5).

### 5.3. NCC Materials Utilized Outside Academia

As shown in Figure 4, general participants in the Nurses Climate Challenge reported 184 educational sessions since the inception of the program (2018). These participants were not faculty (e.g., staff nurses, nurse leaders, or community members), and they did not report educating nurses or nursing students on behalf of a school of nursing. Participants reporting educating others outside of a nursing school setting reached 6250 individuals, with a median audience of 16 individuals per session (mean 34, range 1–450) (see Figure 5). Overall, the number of educational sessions held outside academia comprised 25.4% of the full 724 educational sessions reported via the comprehensive Nurses Climate Challenge campaign between 2018 and 2022.

### 5.4. Impact of Climate and Health Education for Students

While faculty are not required to enter data in the NCC education tracker pertaining to student feedback or results of each educational session, many do. Overall, student feedback was very positive, with multiple comments highlighting how climate change and health information was new to nursing students. Several faculty also shared that students expressed interest in learning about how to become involved. Results of educational sessions varied from classroom discussions to developing student groups or committees to take action on climate change. See Table 2 for examples of qualitative data volunteered by SON Commitment faculty members. See Table 3 for case study reports of faculty and student experience in the NCC.

Throughout the duration of the NCC SON Commitment, multiple faculty and doctoral students have used NCC resources creatively. See two case studies that represent innovative integration of climate change and health.

## 6. Discussion

### A Relatively Small Scope, despite a Widespread Issue

No region of the U.S. will be safe from climate change, and patients and communities will continue to face the brunt of worsening health due to climate impacts. The NCC has engaged tens of thousands of nurses and health professionals (including future generations) and seems to serve as a successful developing framework for nurse-led climate and health education across practice and educational settings. Furthermore, the SON Commitment has brought climate change into the classroom and the clinical setting.

However, it is unclear whether incoming nurses within the profession are widely climate-literate. According to the U.S. Global Change Research Program, individuals with climate literacy understand the essential principles of earth’s climate and can critically assess information about the climate, communicate meaningfully about climate change, and make informed decisions with the climate in mind [40]. Many SON Commitment faculty expressed that climate change is integrated in their nursing school due to personal scholarship or personal interest only. This combined lack of climate literacy with a main driver for climate integration in curricula being driven by personal passions will not be sufficient for adequately reaching the nearly one hundred thousand nursing students graduating annually across the U.S. [41]. In considering why the NCC SON Commitment has not been implemented in greater numbers of nursing schools, the lack of clear language from organizations that establish curricular mandates may be an indicator. In addition, the volume of content nursing faculty are expected to teach is large; this poses further hurdles for faculty, whether or not they may be inclined to include climate change in their courses in the first place. While increasing numbers of universities are beginning to include climate-change adaptation and mitigation in their overall strategic planning [30], the connections between climate change and human health are relatively poorly understood outside of the health care sector. As cited in the American College & University Presidents’ Climate Commitment (ACUPCC) report: “At a basic level, building environmental and climate literacy in all graduates is helping to prepare the citizens and community members of tomorrow by helping them to understand climate change impacts and systems thinking” [42]. The report includes minor references to “health” or “public health,” but none focused on “nursing” [42]. Thus, university leadership may not see climate change as a critical aspect to integrate into nursing schools.

First-year results of the Climate, Health, and Nursing Tool (CHANT) found that, while awareness and concern for climate change is relatively high in the general nursing population, there is a widespread lack of confidence, motivation, and overall climate action in that same population (examples of reported actions included reducing supply and energy usage in a practice setting, decreasing landfill contribution in a practice setting, and communicating with patients, family, and elected officials) [43]. Schenk and colleagues reported that this lack of translating concern into action occurred evenly across respondents, including faculty, students, and practicing nurses [43]. Notably, due to snowball recruitment in the CHANT as well as the NCC SON Commitment, the subset of faculty who have taken the CHANT also likely serve as faculty contacts for the Commitment. Taking this into account, it is possible—perhaps likely—that early adopters to integrating climate and health content into nursing courses still feel a lack of confidence or motivation and are faced with barriers to climate action that discourage a widespread movement.

The timing of this initiative crossed over with the global pandemic COVID-19. It is likely that this influenced participation in the Nurses Climate Challenge by direct care nurses during 2019 and 2020 in particular. Many faculty who taught in schools that were closed to in-person instruction (lectures, labs, clinicals, or a combination) expressed seeking online or hybrid experiences to innovatively fill gaps in their curriculum. This may have contributed to the uptick of faculty utilizing resources in 2019 and 2020. However, by the time faculty were reporting the highest number of educational sessions and students reached (2022), programs had shifted back to in-person content delivery.

If we as a global community are to eliminate fossil fuel emissions by mid-century as directed by the Intergovernmental Panel on Climate Change [44], far more nurses, nursing educators, and nursing students must be prepared to take leadership in climate mitigation and the environmentally sustainable transformation of the health sector in support of climate-smart health care [45]. Personal interest in the integration of climate change into nursing (and all) curricula does not suffice now, and it certainly will not serve our species into the future as we face the increasingly dire effects of ever-rising temperatures on the planet.

## 7. Limitations

While the NCC SON Commitment seems to serve as a beneficial program for nursing students across the U.S. (and potentially beyond), the program has limitations. First, because the NCC SON Commitment does not require a signed contract, faculty and nursing schools are not beholden to their agreement to participate. In some cases, this has impacted the nursing school’s public announcement and promotion of the partnership, utilization of NCC resources, education tracking, and participation in the faculty community of practice.

Further, in the effort to decrease the burden of participation for faculty and increase the likelihood of initiative participation and education reporting, the data entry fields for tracking are limited. Entry fields include activity name (required), presentation date (required), number educated (required), presentation type (not required), presentation setting (not required), audience feedback (not required), and follow-up actions (not required). Many faculty chose not to provide data that was not required, disallowing for broad scale analysis.

While the NCC resources are valuable as a tool for beginning to integrate climate and health content into courses, they are not all-encompassing. Additional focus is needed on climate-smart health care, sustainability, and climate justice, among other climate and health topics.

Finally, beyond gathering data via the online form, the NCC Steering Committee does not verify the actual education delivered by faculty, due to time constraints. Therefore, it is not possible to verify that all of the individuals that were reported to receive education actually received education via the NCC resources.

## 8. Programmatic Expansion

Despite programmatic limitations, the NCC has established roots beyond the original vision of the campaign. In early 2020, Health Care Without Harm-Europe collaborated with ANHE to establish a Nurses Climate Challenge-Europe [46]. Many of the NCC-U.S.’s resources were revised to reflect the climate and health issues specific to Europe’s geography and culture. Shortly after the launch of the initial Challenge (nursing-led education), NCC-Europe established a focus group of European nursing faculty members to participate in the development of Europe’s own NCC SON Commitment as well. As of 2024, 20 schools across Spain, Germany, Malta, Norway, the U.K., Finland, Israel, Greece, Scotland, Portugal, Belgium, Ireland, and Pakistan (outside Europe) have signed onto the NCC-Europe SON Commitment, which is available in seven languages [46]. The NCC-Europe has led to cross-national climate nursing engagement, with discussions of further collaborations regarding the development of an international nursing agenda for climate change research.

While programmatic expansion occurs in Europe, the NCC-U.S. (also inclusive of Canada) arm will continue to grow as well. Teaching resources will expand to include climate-change curricular recommendations based on the 2021 AACN Essentials, the Global Consortium for Climate and Health Education, and other national climate and health curricular guidance, on which ANHE began revision in 2021. In addition, the NCC SON Commitment faculty community of practice is also undergoing process improvements with the aim of expanding support, networking, and resource sharing among faculty. This includes a continued focus on recruitment efforts to expand reach and momentum. Potential topics for future resource development could include more in-depth action-oriented content on climate change mitigation in health care such as decarbonization, as well as resources oriented towards climate resilience.

Though not an established Nurses Climate Challenge program, the single Colombian university that participated in the SON Commitment collaborated with peers in the geographic region to establish ANHE Latinoamérica [47]. AHNE Latinoamérica serves as a promising example of regionalized climate and nursing collaboration and action.

## 9. Conclusions

The Nurses Climate Challenge School of Nursing Commitment may serve as a model for grassroots climate and health integration into health curricula. Faculty who participated in the program had a broad reach, impacted thousands of students, and inspired the future nursing profession to act on climate. Examples of further considerations for research include a study exploring why climate change education has not yet been fully integrated into health professional programming; a deeper analysis of the implementation of the SON Commitment across the participating schools, such as considering how educators facilitated learner engagement with NCC resources; and an investigation into the environmental (greenhouse gas) impact of associated climate actions by learners.

Until national nursing curricular bodies mandate widespread integration of climate change, faculty who independently seek knowledge about the intersection of human health and climate change will continue to serve as the main source of knowledge about how nurses can and must become climate-literate. It is hoped that climate literacy, combined with the understanding of the U.S. health sector’s impact on greenhouse gas emissions, will spur the nursing profession to act in the best interest of our patients—as both individuals and a collective. With a long legacy of holism, innovation, and moral aptitude, it is only fitting that the profession of nursing encompasses climate action.

## Figures and Tables

**Figure 1 ijerph-21-00589-f001:**
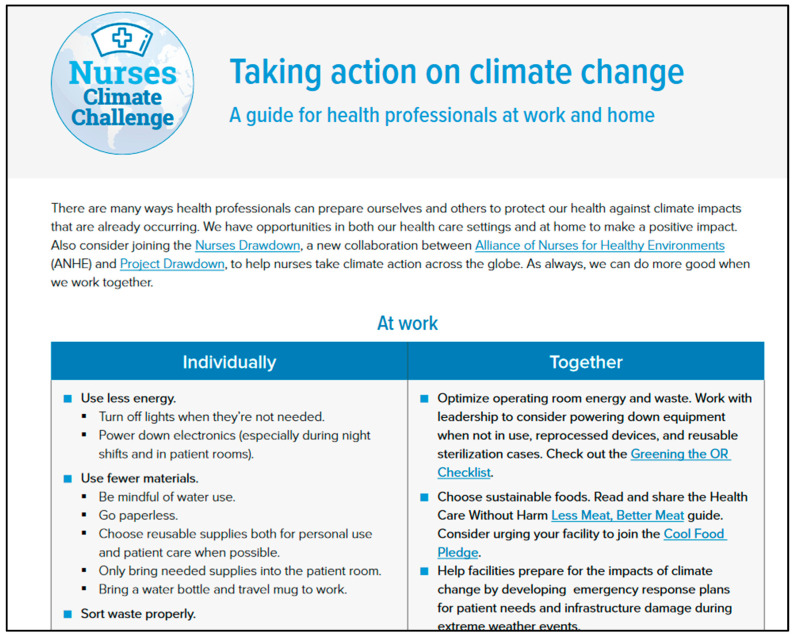
Example of Nurses Climate Challenge resource focused on reducing emissions via conservation of workplace resources.

**Figure 2 ijerph-21-00589-f002:**
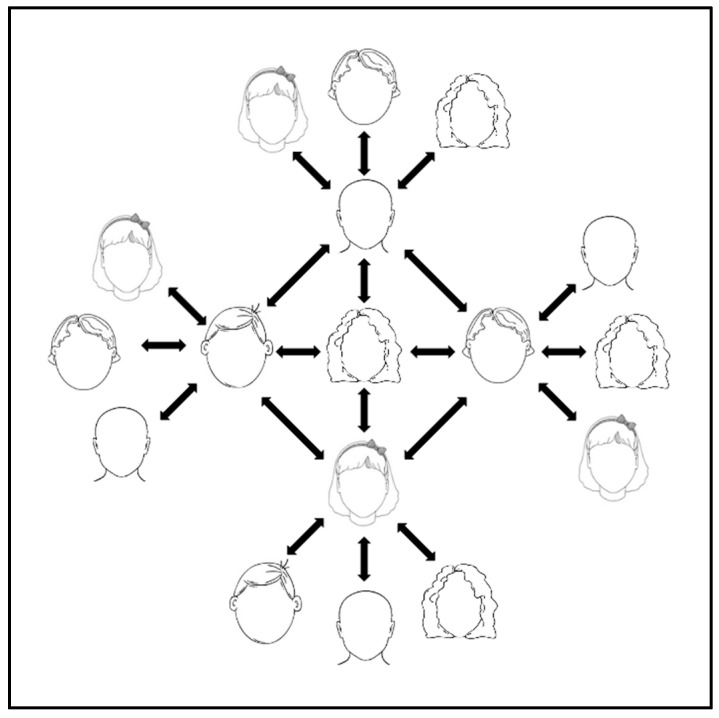
Adapted visualization of the Snowflake Model [36].

**Figure 3 ijerph-21-00589-f003:**
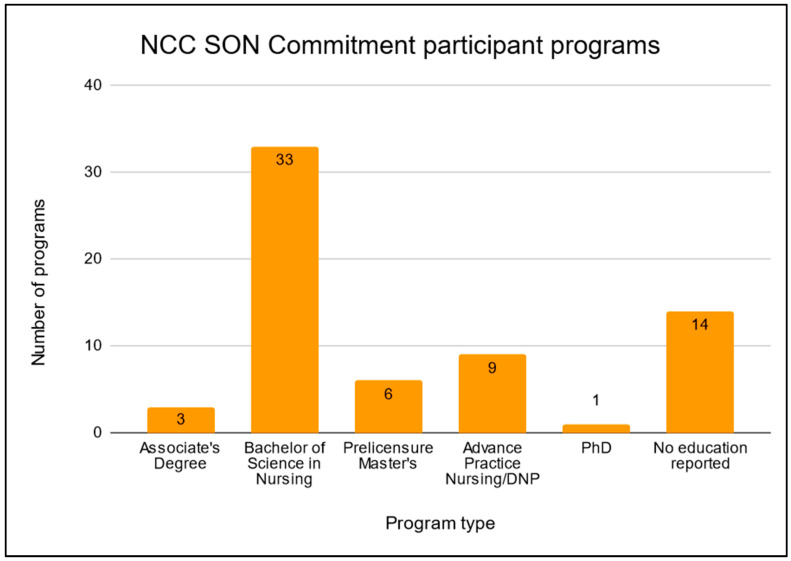
Integration of NCC materials into SON Commitment schools by program type; n = 61, with program overlap amongst 7 participants.

**Figure 4 ijerph-21-00589-f004:**
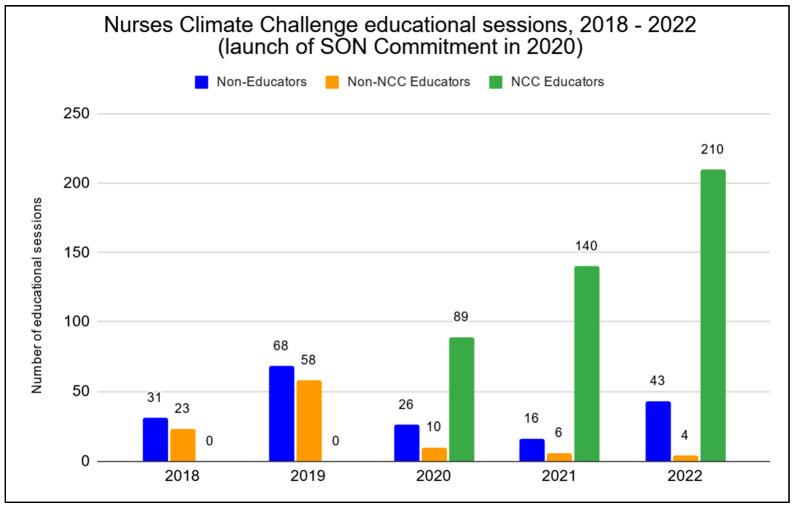
Nurses Climate Challenge educational sessions categorized by non-academic vs. educators not participating in the NCC vs. educators participating in the NCC (2018–2022). Note: the SON Commitment launched in 2020; n = 724.

**Figure 5 ijerph-21-00589-f005:**
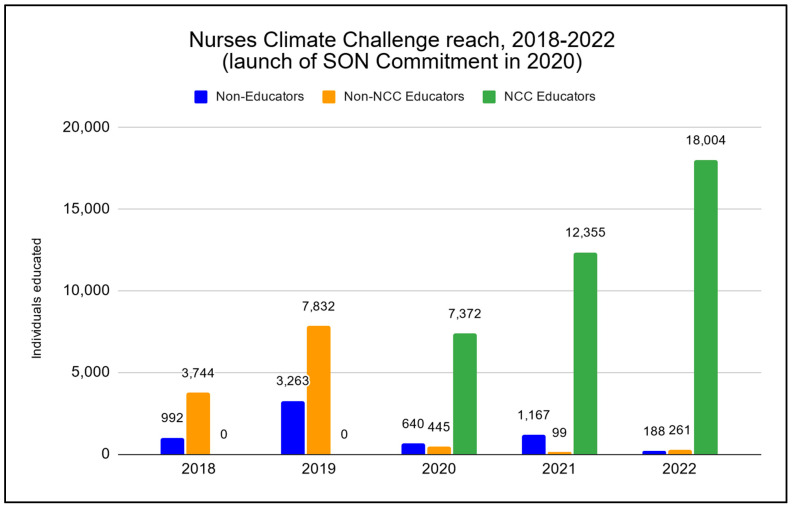
Number of individuals educated by NCC materials separated by audience (2018–2022); n = 56,362.

**Table 1 ijerph-21-00589-t001:** Summary of NCC SON Commitment.

Summary of NCC SON Commitment	
SON agrees to the following:	Promote the NCC to undergraduate and/or graduate nursing students and faculty.
	Use at least one component of NCC content in a course, guest lecture, or presentation.
	Report number of students educated.
NCC agrees to the following:	Display SON name publicly on NCC website.
	Identify SON as participant and highlight individual faculty efforts via social media.
	Publicize select SON faculty and student research on climate and health on NCC website.
	Host community of practice faculty sharing calls prior to fall and spring semesters.
	Provide quarterly reports with number of students educated.

**Table 2 ijerph-21-00589-t002:** Student feedback and result of educational session, per faculty reports.

Student Feedback	Result of Educational Session
“Students often reflect that they were not aware of how nurses can be involved in recognizing, treating, or preventing aspects of climate change—that it was not covered in their nursing education”. “Most students were not aware about the impact of climate change on health. They were motivated to learn more on the topic and engage in education activities in their hospitals to increase awareness about climate change and health”. “Positive feedback on concrete things clinicians can do to address climate change with their patients”. “Want more info on how to help make change (advocacy)”. “Deep ongoing interest and questions about how to become involved”. “Interest in starting a climate change student group”.	“We discussed the role of nurse-led collaborations that address community health impacts related to climate change and how to form local partnerships”. “Class discussion on impact of excessive pandemic PPE on waste stream”. “Students were encouraged to register as Nurse Climate Champions and take action in their work and school settings”. “Five student videos were produced on environmental risk factors that affect the health of the region’s population”. “Working with student affairs to explore student group option”. “Working toward presenting a student initiative for climate actions to Faculty Assembly”. “Resulted in the creation of a Climate Change Committee with health officers and nurses”.

**Table 3 ijerph-21-00589-t003:** NCC SON Commitment: Real-World Examples.

Case study 1. Dr. Lisa Whitfield-Harris, PhD, MBA, RN, Jefferson College of Nursing
Nursing programs focus on nursing fundamentals to improve students’ patient care knowledge and skills; however, during the Nurses Climate Challenge, faculty introduced students to environmental health concerns of under-resourced communities. In one doctoral program, students enrolled in the population health course were exposed to various environmental health topics including air and water quality, lead exposure, and fracking. Working in groups, students assessed their populations’ environmental health in two different zip codes. After completing their needs assessment, they planned and evaluated a program to deploy in both cities. If the program was not effective, then they revised the program and considered other resources to improve its effectiveness. For example, if students noted a high number of children diagnosed with asthma with high absenteeism rates, then they developed an educational program to instruct parents on emergency measures to assist their child. Students used pre-post surveys to understand changes in knowledge and skills to improve the child’s health and decrease school absenteeism. The program was based on available resources in both zip codes. Students gained knowledge on developing and evaluating programs and understanding how to advocate for under-resourced communities. They also improved their skills to request funds and work with key stakeholders in the communities.
Case study 2. Andrea Earley, CRNA, DNP, Hackensack Meridian Health & Jefferson College of Nursing
Environmental sustainability has always been a deep passion of mine. The topics of climate change and environmental sustainability were not a part of my curriculum as both a graduate and undergraduate nurse-in-training. However, my Doctor of Nursing Practice (DNP) project allowed me to explore those passions independently within the clinical setting of anesthesia and create networks among other practitioners who share my passions. I specifically examined the attitudes and awareness of sustainability initiatives with the operating room (OR) and explored the willingness of providers to engage in evidence-based initiatives. Throughout my DNP project implementation, sustainability experts at Hackensack Meridian Health (HMH) were consulted and provided me with background on current sustainability initiatives currently implemented at my clinical site (Jersey Shore University Medical Center) and other hospitals in the HMH network. The negative impacts of unsustainable practices in the OR exacerbate climate change and negatively impact provider safety. I hope the NCC continues to engage practitioners to improve and expand environmental programs.

## Data Availability

Data are contained within the article via nursesclimatechallenge.org.

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
