# Peer review of "School of Nursing Climate Commitment: Nursing Faculty Bring Climate to the Classroom"

_ijerph, 2024, doi:10.3390/ijerph21050589_

Round 1

Reviewer 1 Report

Comments and Suggestions for Authors

please revise according reviewer comments in attached file

Author Response

Reviewer 1

Please add citation on this sentence (Line 41)

Thank you for this suggestion. We have included a reference for this sentence (Lopez-Medina, et al.).

Please explain the meaning of

“expressed” in this context (Line 48)

We have further elaborated on the term “expressed” and have added citations.

Nursing courses, what kind of

course in this situation? (Line 52)

Thank you for this question; we have clarified the types of nursing courses that do not have strong climate change content integrated.

How climate change impact to

human health? Could you mention? (Line 75)

Thank you for this question; we have added examples of how human health is impacted by climate change.

What solutions to solve this

Problem? (Line 184)

The text that the reviewer shared this response for is as follows:

“Upon educating health professional students, faculty submit a limited amount of data for each educational session including the number of individuals educated, audience feedback, and follow-up actions pertaining to climate solutions that arose from the educational session.”

It is not clear from the reviewer’s question about the selected text what “Problem” is being referred to. We have not addressed this portion of the review.

What the meaning of climate

Literate? (Line 277)

Thank you for this great question. We have defined climate literacy and provided a citation.

Please explain about climate

change action in this community. (Line 297)

Thank you for this question. We have added examples of actions from the CHANT survey.

How to implemented this result? (Line 350)

We have further elaborated on the NCC-Europe, including where SON Commitments have been made and potential for future research collaborations and added a citation.

Please add suggestion for future

Research (Line 366)

Thank you kindly for this suggestion. We have put forward additional considerations for research in the conclusion section.

This sentence not clear (Line 373)

Thank you for this comment. We have clarified this sentence.

Please add references (Line 458)

Thank you for this prompt. We have improved citation of the body of the text throughout. 

We have also added multiple new references, which are included in the final section.

Reviewer 2 Report

Comments and Suggestions for Authors

The paper deals with a topic that is relevant, current and innovative for the specific target group. The paper is very well designed in terms of structure and content and everything is explained in a comprehensible manner. It will certainly be of great interest to the reader and will also encourage them to analyze the consequences of climate change on other target groups that may have been previously neglected and to promote them through appropriate (further training) measures.

Author Response

The paper deals with a topic that is relevant, current and innovative for the specific target group. The paper is very well designed in terms of structure and content and everything is explained in a comprehensible manner. It will certainly be of great interest to the reader and will also encourage them to analyze the consequences of climate change on other target groups that may have been previously neglected and to promote them through appropriate (further training) measures.

Thank you kindly for your review.

Reviewer 3 Report

Comments and Suggestions for Authors

I very much admire the underlying aims and intent of the NCC and hope that useful climate change resources continue to be developed and appropriately facilitated across health fields. I also hope that you are able to share the success of your efforts (ie, your evaluation data) in an appropriate publication. However, I am afraid that I do not regard this paper to constitute a research-based article, and thus I cannot recommend it for publication. 

To explain further, I see the paper as a description of a very-worthwhile initiative with some evaluation data documenting implementation. I, however, was expecting an analysis of the implementation of the NCC - perhaps a study exploring why climate change hasn't been integrated in order to check whether it is simply due to a lack of resources or whether it is additionally due to more structural barriers (as is the case in higher education, for example). Another research study might have comprised a detailed analysis of how educators facilitated learner engagement with the resources, and to what extent they were able to prompt discussion, reflection, and action on the part of learners. Alternatively, I might have expected a detailed argument as to what constitutes climate change information relevant to the audience. 

All the above approaches would also have required theoretical framing to contextualise the arguments and enable others outside the field of nursing to learn from this experience. For future analyses of implementation I would suggest looking at the Capabilities Approach as a useful theoretical and methodological tool.

I was interested to see an example of the resources developed. I would have liked much more explanation on how they were developed beyond the short reference to backwards design. Here, I also note that educational research has shown that simply providing learners with information does not automatically change their thinking and practice. That is in order to prompt involvement in the issues and thereafter engage in action it is necessary to facilitate discussions with greater criticality - eg invite discussion, reflect on the issues, contentions, alternative viewpoints. 

I am pleased to learn that many schools of nursing have now adopted the commitment to engage with climate change education, and I very much wish you luck in developing your materials/guidance in the future. I am sorry that I do not see this submission to be appropriate for IJERPH, but I hope that you find more appropriate publication to share and highlight the NCC. 

Author Response

I very much admire the underlying aims and intent of the NCC and hope that useful climate change resources continue to be developed and appropriately facilitated across health fields. I also hope that you are able to share the success of your efforts (ie, your evaluation data) in an appropriate publication. However, I am afraid that I do not regard this paper to constitute a research-based article, and thus I cannot recommend it for publication. 

I am pleased to learn that many schools of nursing have now adopted the commitment to engage with climate change education, and I very much wish you luck in developing your materials/guidance in the future. I am sorry that I do not see this submission to be appropriate for IJERPH, but I hope that you find more appropriate publication to share and highlight the NCC.

[Editorial board’s comments]:

We have relayed your comments to the editorial office. We believe that article is appropriate as the type of manuscript you submit, so we do not recommend you to change the type of article.

We understand that you are concerned about the reviewer's negative comments. After receiving the reviewer's report, we have invited the academic editors to review all the review reports, including Reviewer 3's report. The academic editors held a positive attitude towards this manuscript and feel that this manuscript is worth revising. So we hope that you continue to revise and submit your revision as soon as possible, and we will invite the academic editors again to check the revised revision.

To explain further, I see the paper as a description of a very-worthwhile initiative with some evaluation data documenting implementation. I, however, was expecting an analysis of the implementation of the NCC - perhaps a study exploring why climate change hasn't been integrated in order to check whether it is simply due to a lack of resources or whether it is additionally due to more structural barriers (as is the case in higher education, for example). Another research study might have comprised a detailed analysis of how educators facilitated learner engagement with the resources, and to what extent they were able to prompt discussion, reflection, and action on the part of learners. Alternatively, I might have expected a detailed argument as to what constitutes climate change information relevant to the audience. 

Thank you kindly for these comments. We have suggested these considerations in the Conclusion section and have further elaborated to put forward additional considerations.

All the above approaches would also have required theoretical framing to contextualise the arguments and enable others outside the field of nursing to learn from this experience. For future analyses of implementation I would suggest looking at the Capabilities Approach as a useful theoretical and methodological tool.

Thank you for this comment. If we embark upon future research as suggested, we will explore the Capabilities Approach.

I was interested to see an example of the resources developed. I would have liked much more explanation on how they were developed beyond the short reference to backwards design. Here, I also note that educational research has shown that simply providing learners with information does not automatically change their thinking and practice. That is in order to prompt involvement in the issues and thereafter engage in action it is necessary to facilitate discussions with greater criticality - eg invite discussion, reflect on the issues, contentions, alternative viewpoints. 

Thank you for this comment. We have further elaborated upon the development of the materials and included multiple new citations.
